# Nuclear Receptor Gene Variants Underlying Disorders/Differences of Sex Development through Abnormal Testicular Development

**DOI:** 10.3390/biom13040691

**Published:** 2023-04-19

**Authors:** Atsushi Hattori, Maki Fukami

**Affiliations:** 1Department of Molecular Endocrinology, National Research Institute for Child Health and Development, 2-10-1 Okura, Setagaya, Tokyo 157-8535, Japan; fukami-m@ncchd.go.jp; 2Division of Diversity Research, National Research Institute for Child Health and Development, 2-10-1 Okura, Setagaya, Tokyo 157-8535, Japan

**Keywords:** digenic inheritance, disorders of sex development, genomic structural variation, gonadal development, nuclear receptor, oligogenicity

## Abstract

Gonadal development is the first step in human reproduction. Aberrant gonadal development during the fetal period is a major cause of disorders/differences of sex development (DSD). To date, pathogenic variants of three nuclear receptor genes (*NR5A1*, *NR0B1*, and *NR2F2*) have been reported to cause DSD via atypical testicular development. In this review article, we describe the clinical significance of the *NR5A1* variants as the cause of DSD and introduce novel findings from recent studies. *NR5A1* variants are associated with 46,XY DSD and 46,XX testicular/ovotesticular DSD. Notably, both 46,XX DSD and 46,XY DSD caused by the *NR5A1* variants show remarkable phenotypic variability, to which digenic/oligogenic inheritances potentially contribute. Additionally, we discuss the roles of *NR0B1* and *NR2F2* in the etiology of DSD. *NR0B1* acts as an anti-testicular gene. Duplications containing *NR0B1* result in 46,XY DSD, whereas deletions encompassing *NR0B1* can underlie 46,XX testicular/ovotesticular DSD. *NR2F2* has recently been reported as a causative gene for 46,XX testicular/ovotesticular DSD and possibly for 46,XY DSD, although the role of *NR2F2* in gonadal development is unclear. The knowledge about these three nuclear receptors provides novel insights into the molecular networks involved in the gonadal development in human fetuses.

## 1. Introduction

Gonadal development is a critical step for achieving reproductive abilities in humans. Many spatiotemporally regulated genes are involved in gonadal development [1]. Genetic variants affecting these genes cause disorders/differences of sex development (DSD), congenital conditions in which the chromosomal, gonadal, or anatomical sex is atypical [2]. In some cases, DSD with aberrant gonadal development is associated with atypical karyotypes such as 45,X/46,XY or 46,XX/46,XY. Variants in many genes have been identified in 46,XY testicular dysgenesis [3,4,5,6]. In contrast, the etiologies underlying testicular development with the 46,XX karyotype (46,XX testicular/ovotesticular DSD) and ovarian development with the 46,XY karyotype have not been well elucidated. Known genetic variants associated with 46,XX testicular/ovotesticular DSD include *SRY* translocations [4,5]; copy number variants involving *FGF9*, *NR0B1*, *NR2F2*, *SOX3*, *SOX9*, *SOX10*, and *SPRY2* [4,7,8,9,10]; and sequence variants in *NR2F2*, *NR5A1*, *RSPO1*, *SOX9*, *WNT2B*, *WNT4*, and *WT1* [4,5,11,12,13,14,15,16]. Variants of *CBX2*, *DMRT1*, and *SOX9* have been reported in 46,XY patients with ovaries or ovotestis [17,18,19]. This review focuses on three nuclear receptors variants, *NR5A1*, *NR0B1*, and *NR2F2,* of which cause DSD through abnormal testicular development. We first, briefly, describe the process of testicular development in humans, provide an overview of the nuclear receptors, and then discuss each of the three nuclear receptors in detail.

## 2. Testicular Development in Humans

In humans, the genital ridge, which gives rise to the testis or ovary, develops on the ventromedial surface of the mesonephros at four weeks of gestation (Figure 1) [1,20]. Genes vital for this bipotential (or indifferent) gonad formation include *WT1*, *NR5A1*, *CBX2*, *LHX9*, *EMX2*, and *GATA4* [21]. At five weeks of gestation, bipotential primordial germ cells in the yolk sac migrate into the developing gonad [1], and coelomic epithelial cells within the genital ridge differentiate into supporting cell precursors (differentiating into Sertoli cells or granulosa cells) and steroidogenic cell precursors (differentiating into Leydig cells or theca cells) [20]. Male gonadal sex differentiation begins in somatic cells, first occurring in supporting cells and subsequently in steroidogenic cells. At six weeks of gestation, *SRY* expression initiates the differentiation of supporting cell precursors into Sertoli cells [1,21]. This male sex differentiation of supporting cells involves *SRY*, *SOX9*, *NR5A1*, *FGF9*, and *PGD2* [1,20,21]. Sertoli cells secrete paracrine factors such as desert hedgehog and platelet-derived growth factor, which drive the differentiation of steroidogenic cell precursors into fetal Leydig cells [1]. Fetal Leydig cells are capable of producing testosterone at seven weeks of gestation [1]. The differentiation of germ cells depends on the Sertoli cells and the Leydig cells. Sertoli cells are crucial for seminiferous tubule formation, which occurs at seven to eight weeks of gestation [1]. Around this time, primordial germ cells differentiate into testicular gonocytes in seminiferous tubules [22]. The differentiation from gonocytes to prespermatogonia occurs asynchronously from approximately 16 weeks of gestation and is supported by the paracrine factors from the Sertoli cells and the Leydig cells [1].

## 3. Overview of Nuclear Receptors

Nuclear receptors belong to a family of transcription factors. Typical nuclear receptors are characterized by a DNA-binding domain and ligand-binding domain, although *NR0B1* and *NR0B2* lack conventional DNA-binding domain [28,29]. The human genome encodes 48 nuclear receptors [28]. These receptors are phylogenetically classified into seven subfamilies, numbered from zero to six [28]. Each subfamily is further classified into alphabetically named groups containing several members. Most nuclear receptors can be classified into four classes according to their modes of action [28]. Class I is composed of steroid receptors. Class I nuclear receptors bind to heat shock proteins in the cytoplasm when unliganded. Ligand binding releases receptors from the heat shock protein, allowing homodimerization. The homodimerized nuclear receptors enter the nucleus and function as transcription factors by binding to inverted repeats of the consensus motifs. Class II nuclear receptors dimerize primarily with retinoid X receptors and bind to DNA regardless of the ligand status. The unliganded and liganded receptors recruit corepressors and coactivators, respectively. Thus, class II nuclear receptors repress gene transcription in the unliganded state and activate transcription in the liganded state. Class III nuclear receptors homodimerize like class I receptors but bind to direct repeats of the consensus motifs. Class IV nuclear receptors typically act as monomers and bind to single consensus motifs rather than to repeats. Nuclear receptors control various biological processes, such as development, metabolism, and reproduction [28]. Among the nuclear receptors, *AR*, *ESR1*, *ESR2*, *NR0B1*, *NR2F2*, *NR5A1*, and *RARA* have been linked to DSD (Table 1) in the Human Genome Mutation Database (HGMD, https://www.hgmd.cf.ac.uk/ac/index.php; accessed on 6 March 2023) or Online Mendelian Inheritance in Man (https://www.omim.org; accessed on 6 March 2023).

## 4. *NR5A1* (Nuclear Receptor Subfamily 5 Group A Member 1)

*NR5A1* (HGNC ID, HGNC:7983; chromosomal location, 9q33.3) encodes steroidogenic factor 1 (SF-1, NP_004950, alias, adrenal 4-binding protein), which is a critical factor in adrenal, gonadal development, and steroidogenesis [30,31]. The ligands for the NR5A1 protein have not yet been identified, although phospholipids have been reported as candidates [30]. *NR5A1* is primarily expressed in the adrenal glands, testes, ovaries, hypothalamus, pituitary gland, spleen, skin, and uterus [30]. In the testes, *NR5A1* is expressed in Sertoli cells and Leydig cells throughout fetal development and postnatal life [23,24]. *NR5A1* functions at various stages of testicular development. *NR5A1* likely contributes to the formation of the genital ridge because the genital ridge expresses *NR5A1* [23] and fails to develop in *NR5A1* knockout mice [32]. In the first step of male sex differentiation, supporting cell precursors require *NR5A1* to differentiate into Sertoli cells. In cooperation with SRY and SOX9, NR5A1 activates the transcription of *Sox9*, which drives the differentiation of supporting cell precursors into Sertoli cells [33]. *Amh* is another target gene of NR5A1 in Sertoli cells [30]. *Amh* encodes anti-Müllerian hormone, which facilitates Müllerian duct regression. NR5A1 is also expressed in fetal Leydig cells, activating the transcription of steroidogenic genes such as *Star*, *Cyp11a1*, *Cyp17a1*, and *HSD3B2* [30,31]. Reflecting the critical role of *NR5A1* in testicular development and androgen production, pathogenic variants of *NR5A1* cause 46,XY DSD via testicular dysgenesis and dysfunction.

### 4.1. NR5A1 Variants in 46,XY DSD

In 1999, sequencing of *NR5A1* in one patient showed for the first time that *NR5A1* variants caused 46,XY DSD and primary adrenal insufficiency [34]. Subsequent studies revealed that the heterozygous variants in *NR5A1* are one of the most frequent causes of 46,XY DSD [6]. Previous studies have suggested that 2.2–15.4% of patients with 46,XY DSD have rare *NR5A1* variants that likely contribute to the phenotype [35,36,37,38,39,40,41]. According to the Human Gene Mutation Database, more than 200 *NR5A1* variants are associated with 46,XY DSD (Table 1). Although most patients have single nucleotide substitutions or small indels, some have structural variants (genetic alterations greater than 50 bp) involving the exons of *NR5A1* [42,43]. *NR5A1* variants do not affect adrenal functions in most cases; however, several variants have been identified in patients with primary adrenal insufficiency [34,44,45]. In DSD, *NR5A1* variants result in broad phenotypes ranging from male- to female-type external genitalia [45]. Notably, most patients with *NR5A1* variants show spontaneous virilization during puberty [45], which should be considered during sex assignments and surgical procedures. In one study that investigated the gender of 46,XY DSD patients with *NR5A1* variants over time, all six patients with *NR5A1* variants and female gender assignment at birth were reassigned as male afterward, while none of the eight patients with male gender assignment at birth received gender reassignment [46]. Therefore, the correct diagnosis of *NR5A1* abnormalities improves the management of patients with 46,XY DSD.

### 4.2. NR5A1 Variants in 46,XX Testicular/Ovotesticular DSD

*NR5A1* variants also cause 46,XX testicular/ovotesticular DSD. Bashamboo et al. [12], Baetens et al. [13], and Igarashi et al. [14] independently identified a heterozygous variant of *NR5A1* (c.274C>T, p.Arg92Trp) in patients with 46,XX testicular/ovotesticular DSD. Previous studies have identified 16 cases of 46,XX testicular/ovotesticular DSD associated with the heterozygous p.Arg92Trp variant [12,13,14,47,48,49,50], as well as two cases with the c.275G>A (p.Arg92Gln) or c.779C>T (p.Ala260Val) variants [49,51]. Meanwhile, three 46,XY individuals with the heterozygous p.Arg92Trp [12,52] and homozygous p.Arg92Gln variants [44] have been reported to have gonadal dysgenesis. It is intriguing that the two variants exerted paradoxical effects on genetic males and females, namely, testicular development in the 46,XX individuals (a “pro-testicular” effect) and testicular dysgenesis in the 46,XY individuals (an “anti-testicular” effect). The mechanisms underlying this phenomenon have not yet been fully elucidated. As an explanation for the mechanisms underlying 46,XY DSD, Bashamboo et al. showed that the p.Arg92Trp mutant protein failed to bind an NR5A1-binding sequence (CCAAGGTCA) and had a reduced ability to activate TESCO (a testis-specific enhancer for *Sox9*) and the promoters of *AMH* and *Cyp11a1* [12]. Furthermore, two studies using luciferase assays have suggested explanations for testicular development in the 46,XX individuals. First, Bashamboo et al. showed that the p.Arg92Trp mutant failed to synergize with the β-catenin to activate *Nr0b1* promoter (Figure 2) [12]. Second, Igarashi et al. showed that the activation of TESCO by the p.Arg92Trp mutant was not inhibited by NR0B1 [14]. In contrast to human phenotypes, XX mice harboring homozygous and heterozygous p.Arg92Trp variants of *Nr5a1* do not develop testicular tissues [53]. Consistent with this finding, *Nr5a1* expression is repressed in the developing ovary of mice [54] and rats [55], whereas *NR5A1* is expressed in human ovaries at 6–10 weeks of gestation [12]. Similar to the p.Arg92Trp variant, the p.Ala260Val variant may cause 46,XX ovotesticular DSD through an alteration of synergy with the β-catenin on the transcriptional regulation of *NR0B1* [49]. The mechanism underlying 46,XX ovotesticular DSD associated with p.Arg92Gln [51] remains to be investigated.

### 4.3. Findings Obtained from Recent Studies

#### 4.3.1. Potential Contribution of Digenic/Oligogenic Inheritance to the Broad Phenotypic Spectrum Associated with NR5A1 Variants

*NR5A1* variants in 46,XY patients are associated with a broad phenotypic spectrum, ranging from female phenotypes to isolated hypospadias or male infertility. This phenotypic variability was also observed in members of a family sharing the same variant [56] and even in dizygotic twins who were supposed to be exposed to similar environments during the fetal period [57]. Likewise, the phenotypes of the 46,XX individuals associated with p.Arg92Trp varied from 46,XX testicular DSD to the typical female phenotype [12,13,50].

One possible explanation for this phenotypic variability is digenic or oligogenic inheritance. In 46,XY DSD patients with variants in *NR5A1*, additional variants that potentially modify phenotypes have been identified in at least 37 genes [35,58,59,60,61,62,63,64,65,66]. Digenic inheritances are the simplest among the digenic/oligogenic inheritances and have occasionally been identified in a manner similar to the Mendelian inheritance [67]. Among the 37 genes mentioned above, genes that potentially contribute to digenic inheritances in combination with *NR5A1* include *AMH*, *AR*, *FLRT3*, *INHA*, *MAP3K1*, *SOX3*, *STAR*, *SRY*, and *ZFPM2* [35,61,62,63,64,65,66]. In many cases of digenic inheritance, proteins encoded by the two genes have protein-protein interactions [68]. In this regard, SRY is highly likely to have protein-protein interactions with NR5A1 [33]. Wang et al. reported a patient with 46,XY DSD who had a heterozygous variant of *NR5A1* (p.Gly212Ser) and a hemizygous variant of *SRY* (p.Arg76Leu) [65]. The patient had female-type external genitalia and her chief complaint was primary amenorrhea. Ultrasonography identified no gonads. The p.Gly212Ser variant of *NR5A1* was assessed as pathogenic, whereas the p.Arg76Leu variant was classified as likely pathogenic according to the ACMG guidelines [65,69]. The heterozygous p.Gly212Ser variant of *NR5A1* has been identified in a man with infertility, and therefore, likely exerts only mild effects on testicular development and function [70]. Considering that SRY cooperates with NR5A1 to upregulate the transcription of *Sox9* [33], the p.Arg76Leu variant in *SRY* may have modified the phenotype caused by the pathogenic p.Gly212Ser variant in *NR5A1*, contributing to severe undermasculinization in this patient.

Similarly, AR has been reported to interact with NR5A1. In the study by Wang et al., a patient with 46,XY DSD was reported to have a heterozygous variant of *NR5A1* (p.Thr29Lys) and a hemizygous variant of *AR* (p.Leu295Pro) [65]. The patient presented with a micropenis, perineal hypospadias, bilateral cryptorchidism, and a bifid scrotum. Both variants were classified as likely pathogenic according to the ACMG guidelines [65,69]. Pull-down assays revealed a physical interaction between NR5A1 and the DNA-binding domain of AR [71]. While the central mechanism of DSD caused by *AR* variants is an alteration of the extragonadal response to androgens, aberrations in testicular development or function may play a role because *AR* is expressed in Leydig cells of developing human testes [72]. O’Shaughnessy et al. suggested a potential role of *Ar* in Leydig cells by comparing the responses of Leydig cells to human chorionic gonadotropin in gonadotropin-deficient mice with and without *Ar* knockout [73]. The authors showed that *Ar* knockout diminished the Leydig cell proliferation and downregulated the expression of Leydig cell-specific genes, such as *Lhr*, *Cyp17a1*, *Hsd3b6*, *Hsd17b3*, and *Insl3* [73]. Notably, *Ar* knockout mice exhibited gonadotropin deficiency, which is not observed in patients with androgen insensitivity syndrome (AIS). Patients with AIS occasionally exhibit high levels of gonadotropin and testosterone [74,75]. High luteinizing hormone levels in patients with AIS potentially compensate for the adverse effects of the *AR* variants on the Leydig cells if patients do not have additional variants in other genes. However, we cannot exclude the possibility that *AR* variants, when combined with pathogenic variants in other genes, may affect Leydig cell development and function to an extent that cannot be compensated.

#### 4.3.2. Variants in Regulatory Regions of NR5A1 as a Potential Etiology of DSD

Given that the expression of transcription factors, such as *SRY*, *SOX9*, *NR5A1*, and *WT1*, is strictly regulated both spatially and temporally, it is natural to assume that non-coding variants altering gene expression may contribute to the etiology of DSD. As for *SOX9*, chromosomal amplifications involving a 68 kb region 516 kb upstream of *SOX9* cause 46,XX testicular/ovotesticular DSD, whereas deletions involving a 33 kb region 607 kb upstream of *SOX9* cause 46,XY gonadal dysgenesis [76,77]. Regarding *NR5A1*, Fabbri-Scallet et al. reported four non-coding variants in three patients [78]. Among these variants, a combination of two heterozygous variants (c.-413G>A and c.-207C>A) in one patient showed reduced promoter activity in the luciferase assays. Importantly, c.-413G>A and c.-207C>A were localized within SP1 and WT1 binding sites, respectively. In mice, a fetal Leydig cell-specific enhancer was identified at a 3.1 kb upstream of *Nr5a1* [79]. The variants in the region of the human genome homologous to this murine enhancer may cause DSD, although such variants have not yet been identified in patients with DSD.

## 5. *NR0B1* (Nuclear Receptor Subfamily 0 Group B Member 1)

*NR0B1* (HGNC ID, HGNC:7960; chromosomal location, Xp21.2) encodes a protein (NP_000466), also known as the dosage-sensitive sex reversal-adrenal hypoplasia congenita critical region on the X chromosome protein 1 (DAX1). *NR0B1* is primarily expressed in the adrenal gland, testes, ovaries, and pituitary gland [80]. During testicular development, *NR0B1* expression is detected in the genital ridge at five weeks of gestation and persists after sex determination [25]. At seven weeks of gestation, *NR0B1* expression overlies the distribution of Sertoli cells in the seminiferous tubules [25]. NR0B1 is an orphan nuclear receptor with no known ligand identified. NR0B1 lacks the conventional DNA-binding domain and instead harbors N-terminal 3.5 repeats of a 65–67 amino acid motif containing two putative zinc fingers within each motif [80]. Although the biological roles of NR0B1 are unclear, NR0B1 is thought to regulate gene transcription by interacting with other nuclear receptors such as NR5A1 and NR5A2 [45], and/or by directly binding to the DNA [80]. Reporter assays in cultured cells suggested that NR0B1 downregulated the promoters and enhancers of genes required for testicular development and androgen synthesis, such as *SOX9*, *STAR*, *CYP11A1*, *CYP17A1*, and *HSD3B2* [81,82]. Congenital adrenal hypoplasia and hypogonadotropic hypogonadism are the most well-known phenotypes caused by *NR0B1* variants [45].

### 5.1. Copy Number Variants around NR0B1 in 46,XY DSD

Phenotypes associated with copy number variants encompassing *NR0B1* suggest that *NR0B1* is an anti-testicular gene. Duplications involving *NR0B1* have been reported in several patients with 46,XY DSD (Figure 3) [83,84,85,86,87,88,89,90,91,92,93]. Most patients presented with gonadal dysgenesis, whereas one patient had ovarian tissues [85]. Bardoni et al. identified a dosage-sensitive sex reversal (DSS) region of 160 kb, which represented the minimal overlapping region of duplications in patients with 46,XY DSD [86]. The DSS region contains candidate genes for 46,XY DSD (four *MAGEB* genes and *NR0B1*) [83,94]. Subsequent studies narrowed the region responsible for the 46,XY DSD to approximately 70 kb [83,84,85,87,89,92]. Especially, Dong et al. reported a duplication that did not contain the *MAGEB* genes [87]. This duplication provides evidence supporting *NR0B1* as a candidate gene for 46,XY gonadal dysgenesis.

While the above-mentioned copy number variants encompass the *NR0B1* gene body, Xp21.2 copy number variants outside *NR0B1* have also been associated with 46,XY DSD (Figure 3). A 250 kb deletion 11 kb upstream of *NR0B1* was identified in a 46,XY patient who had gonadal dysgenesis without seminiferous tubules or ovarian follicles [95]. Although the potent enhancer 4 kb upstream of *NR0B1* was intact, the deleted regions contained a cluster of NR5A1 binding sequences and evolutionarily conserved segments [95]. Thus, the deletion may have altered *NR0B1* expression. Recently, duplications containing a potential enhancer element have been identified in two individuals with 46,XY DSD [96,97]. This enhancer element resides in a topologically associating domain (TAD) that contains *TASL* and *GK* in the male control genome, but not *NR0B1* [96]. In contrast, genomic rearrangements in one of the two patients disrupted conventional TAD, creating an aberrant interaction between *NR0B1* and the enhancer element [96]. Although whether the duplication in another patient affects the TAD around *NR0B1* remains to be investigated [97], the duplication might have altered the spatial relationship between *NR0B1* and this putative enhancer or other regulatory elements, upregulating the transcription of *NR0B1*.

**Figure 3 biomolecules-13-00691-f003:**
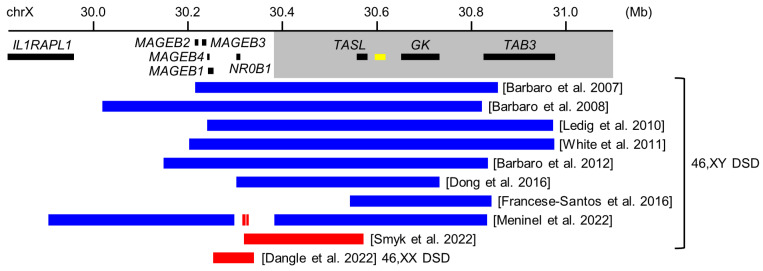
Copy number variants around *NR0B1* underlying disorders/differences of sex development (DSD). Genomic positions are described according to the GRCh38/hg38 reference genome. The yellow bar depicts the potential enhancer proposed by Meniel et al. [96]. This enhancer resides in a topologically associating domain (a region shadowed in gray) that is separated from *NR0B1.* Blue and red lines depict duplications and deletions, respectively. A genomic rearrangement reported by Meninel et al. [96] contains an inversion of a region between two deletions. The copy number variants reported by Francese-Santos et al., Meninel et al., and Smyk et al. [95,96,97] do not involve the *NR0B1* gene body. Barbaro et al. 2007 [83]; Barbaro et al. 2008 [84]; Ledig et al. 2010 [89]; White et al. 2011 [92]; Barbaro et al. 2012 [85]; Dong et al. 2016 [87]; Dangle et al. 2022 [8].

### 5.2. NR0B1 Sequence Variants in 46,XY DSD

Importantly, the hypogonadotropic hypogonadism associated with *NR0B1* can cause clinical signs compatible with DSD, such as micropenis and cryptorchidism. One patient with the hemizygous p.Trp171* variant in *NR0B1* presented with hypospadias, micropenis, bilateral cryptorchidism, and no Müllerian derivatives [98]. Most of his clinical signs could be explained by hypogonadotropic hypogonadism, although the hypospadias was atypical. Another patient with hypospadias was reported to harbor a hemizygous p.Tyr91* variant of *NR0B1* [87]. To our knowledge, no single nucleotide substitutions or small indels in *NR0B1* have been identified in 46,XY individuals with obvious testicular dysgenesis.

Overall, the hypogonadotropic hypogonadism associated with *NR0B1* variants appears to be relatively mild and micropenis at birth is rare in individuals with *NR0B1* variants. Puberty starts spontaneously in some patients with *NR0B1* variants but is usually arrested before completion [45]. Small penile size in adulthood, cryptorchidism, and gynecomastia due to hypogonadism are occasionally present [99,100].

*NR0B1* variants can cause infertility in men without the apparent hypogonadotropic hypogonadism. Specifically, a hemizygous c.965C>G (p.Thr322Ser) variant was identified in an infertile man [101]. Another man with the hemizygous p.Trp39* variant in *NR0B1* experienced spontaneous puberty but presented with oligospermia [102]. A man with the hemizygous p.Gly169Alafs*95 variant of *NR0B1* was reported to have Sertoli cell-only syndrome [103]. In addition, the genetic sequencing of 776 patients with azoospermia identified six missense variants in *NR0B1* that were absent in 709 fertile men [104].

### 5.3. Copy Number Variants around NR0B1 in 46,XX Ovotesticular DSD

Dangle et al. identified an 80 kb deletion encompassing *NR0B1* and putative *MAGEB* regulatory regions in a patient with 46,XX ovotesticular DSD (Figure 3) [8]. The authors speculated that the combination of a one-copy loss of anti-testicular *NR0B1* and overexpression of pro-testicular *MAGEB* genes resulted in testicular development in the patient.

### 5.4. Roles of NR0B1 in Sexual Development: Implications from Studies in Rodents

Studies in 46,XY mice have suggested that overexpression of *Nr0b1* contributes to defects in testicular development, whereas deletion of *Nr0b1* causes various phenotypes ranging from spermatogenic failure to complete sex reversal. Ludbrook et al. demonstrated that *Nr0b1* overexpression reduces *Sox9* expression in male mice [82]. Furthermore, the combination of heterozygous *Sox9* knockout and the overexpression of *Nr0b1* led to the formation of ovotestes in XY mice [82]. However, the results of *Nr0b1* deletion appeared to depend on the strain. For example, the deletion of exon 2 of *Nr0b1* in male 129Sv/J mice disrupted spermatogenesis, whereas *Nr0b1* deletions in male mice with mixed genetic backgrounds led to complete sex reversal [105,106,107]. Interestingly, *Nr0b1* knockout may result in excessive androgen production. Leydig cell-specific *Nr0b1* knockout mice have a higher testosterone levels than those in wild-type mice during the first three to four weeks after birth [108]. The expression levels of steroidogenic genes (*Star*, *Cyp11a1*, *Cyp17a1*, *and Hsd3b1*) also increased in the testes of *Nr0b1* knockout mice [108]. Consistent with this finding, some patients with peripheral precocious puberty also harbored *NR0B1* variants [109,110]. Furthermore, multiple patients with *NR0B1* have been reported to experience central precocious puberty [111,112,113] although the underlying mechanism remains to be elucidated.

## 6. *NR2F2* (Nuclear Receptor Subfamily 2 Group F Member 2)

*NR2F2* (HGNC ID, HGNC:7976; chromosomal location, 15q26.2) encodes chicken ovalbumin upstream promoter transcription factor 2 (COUP-TF2, NP_066285). The NR2F2 protein is an orphan nuclear receptor as no ligands are known. Although *NR2F2* is widely expressed, it is primarily expressed in mesenchymal cells during fetal development [114] and plays a critical role in mesoderm formation [115]. Consistent with the expression pattern of *NR2F2*, congenital heart defects are the most well-known phenotypes associated with the pathogenic variants of *NR2F2* [114]. In developing human testes, *NR2F2* expression is observed in the Leydig cells from seven to ten weeks of gestation, but is downregulated at 15 weeks of gestation and remains repressed throughout fetal life [26,116]. Previous studies suggest that this *NR2F2* repression is necessary for fetal Leydig cell differentiation [116]. In the adult testes, on the other hand, *NR2F2* is expressed in adult Leydig cells [117]. It is unknown whether *NR2F2* is expressed in fetal Leydig cells in the adult testes. The findings described above indicate that the functions of *NR2F2* are different between fetal and adult Leydig cells, although the precise roles of *NR2F2* in the Leydig cell differentiation remain to be elucidated. According to the rodent studies, the target genes of NR2F2 in the Leydig cells may include *INSL3*, *AMHR2*, and various genes encoding steroidogenic enzymes [116,118,119,120,121].

### 6.1. NR2F2 Variants in 46,XX Testicular/Ovotesticular DSD

In total, four cases of 46,XX testicular/ovotesticular DSD associated with the loss of function variants of *NR2F2* suggest that *NR2F2* is an anti-testicular gene. The first three unrelated cases with the frameshift variants were reported by Bashamboo et al., and one additional case with a 3 Mb deletion encompassing *NR2F2* was described by Carvalheira et al. [9,11]. All four patients had 46,XX karyotypes with inappropriate testicular development. The phenotypes of these patients ranged from ambiguous to male-type genitalia. Other characteristic features shared by some patients included various cardiac defects, congenital diaphragmatic hernia, blepharophimosis, ptosis, and epicanthus inversus. The mechanism underlying testicular development associated with *NR2F2* variants remains unclear.

### 6.2. NR2F2 Sequence Variants in 46,XY DSD

Zidoune et al. identified a de novo heterozygous missense variant (c.737G>A, p.Arg246His) in *NR2F2* in a patient with 46,XY DSD [62]. The patient was a boy with a micropenis, middle hypospadias, palpable gonads in a well-developed scrotum, and no Müllerian structures. A human chorionic gonadotropin stimulation test performed at two years of age showed sufficient testosterone production. He had two additional rare heterozygous variants in the candidate genes for DSD (p.Pro1554Leu in *GLI2* and p.Met1312Arg in *GLI3*). Although the p.Arg246His variant in *NR2F2* may cause DSD, the patient’s clinical signs suggested normal testicular function. Thus, whether the *NR2F2* variants cause 46,XY DSD through testicular dysgenesis remains unknown.

### 6.3. Roles of NR2F2 in Sexual Development: Implications from Studies in Rodents

*Nr2f2* seems to be necessary for Leydig cell development in rodents. Similar to humans, *NR2F2* is expressed in the interstitial regions of the testes in rodents [122]. *Nr2f2* knockout in prepubertal male mice disrupted adult Leydig cell development and spermatogenesis [123]. The *Nr2f2* expression in rat Leydig cells decreases during fetal maturation and is inversely correlated with the expression of steroidogenic genes (*Star*, *Cyp11a1*, *Cyp17a1*, and *Hsd3b1*). The promoter regions of these genes contain overlapping binding sites for NR5A1 and NR2F2 [116]. Thus, NR5A1 and NR2F2 may compete for the regulation of their target genes. The inverse expression of *Nr2f2* and steroidogenic genes may reflect the role of *Nr2f2* in preserving pools of Leydig progenitor cells by repressing the functional maturation of the fetal Leydig cells [124,125]. Target genes of NR2F2 may include *Star*, *Cyp11a1*, *Cyp17a1*, *Akr1c14* (murine ortholog of human *AKR1C1*, *AKR1C2*, and *AKR1C3*), *Insl3*, and *Amhr2* [116,118,119,120,121]. In transcriptional regulation of *Insl3* in mice, NR2F2 is believed to bind to a sequence upstream (−97 bp–−83 bp) of *Insl3* and cooperate with NR5A1 to activate transcription [126,127]. While *Nr2f2* probably plays a role in fetal Leydig cell functions, *Nr2f2* knockout in adult mice did not affect Leydig cell numbers and functions, suggesting that *Nr2f2* is dispensable for maintaining adult Leydig cells [123].

## 7. Future Perspective: A New Model to Investigate Molecular Networks in Human Testicular Development

Recently, Gonen et al. established Sertoli-like cell lines derived from induced pluripotent stem cells (iPSCs) of a male individual with the 46,XY karyotype, a female individual with the 46,XX karyotype, and a 46,XY DSD patient with a heterozygous pathogenic variant (p.Arg313Cys) of *NR5A1* [128]. Although both normal 46,XY cells and 46,XY DSD cells expressed *SOX9*, 46,XY DSD cells failed to express the Sertoli marker genes (*GATA4*, *NR5A1*, *FGF9*, and *DMRT1*) to sufficient levels [128]. Moreover, the 46,XY DSD cells aberrantly expressed *FOXL2*, a pro-ovary gene [128]. These aberrant gene expressions were partially recovered by correcting the pathogenic variant using CRISPR-Cas9 genome editing [128]. Although this model may be resource- and time-consuming, it is an attractive tool for evaluating the pathogenicity of variants and for further understanding the molecular mechanisms underlying human testicular development.

## 8. Conclusions

*NR5A1* plays a critical role in testicular development. *NR5A1* variants by themselves, or potentially in combination with variants in other genes, are responsible for various types of DSD. Accumulating data suggest that *NR0B1* and *NR2F2* contribute to sex development, although the precise biological roles of these two genes are largely unknown. Further investigations of these three nuclear receptors will provide novel insights into the molecular networks involved in gonadal development in human fetuses.

## Figures and Tables

**Figure 1 biomolecules-13-00691-f001:**
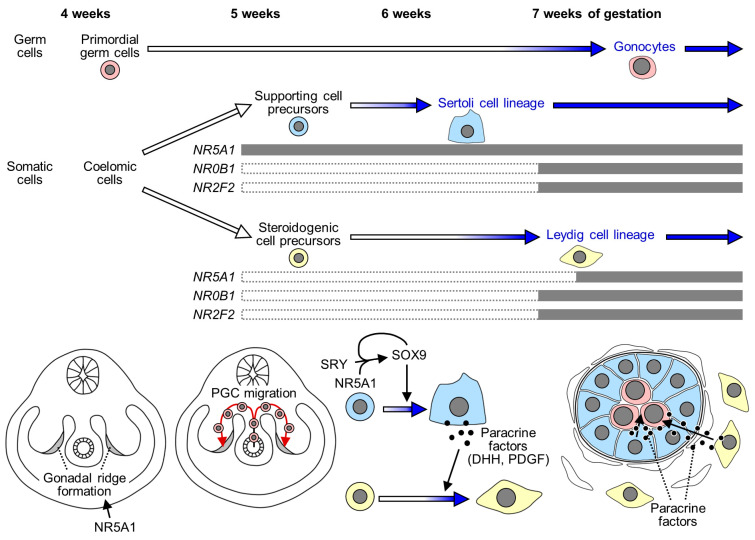
Gonadal differentiation in males. Cells written in blue letters are sexually differentiated (male). Thick arrows depict cell differentiation. Gray boxes indicate gene expression in the cells. Dotted boxes indicate that it is unclear whether the gene is expressed. Gene expression data are based on those published previously [23,24,25,26,27]. Thin black arrows indicate the action of molecules, and thin red arrows depict cell migration. DHH, desert hedgehog; *NR0B1*, nuclear receptor subfamily 0 group B member 1; *NR2F2*, nuclear receptor subfamily 2 group F member 2; *NR5A1*, nuclear receptor subfamily 5 group A member 1; PDGF, platelet-derived growth factor; PGC, primordial germ cell; SOX9, sex-determining region Y-box 9; SRY, sex-determining region Y.

**Figure 2 biomolecules-13-00691-f002:**
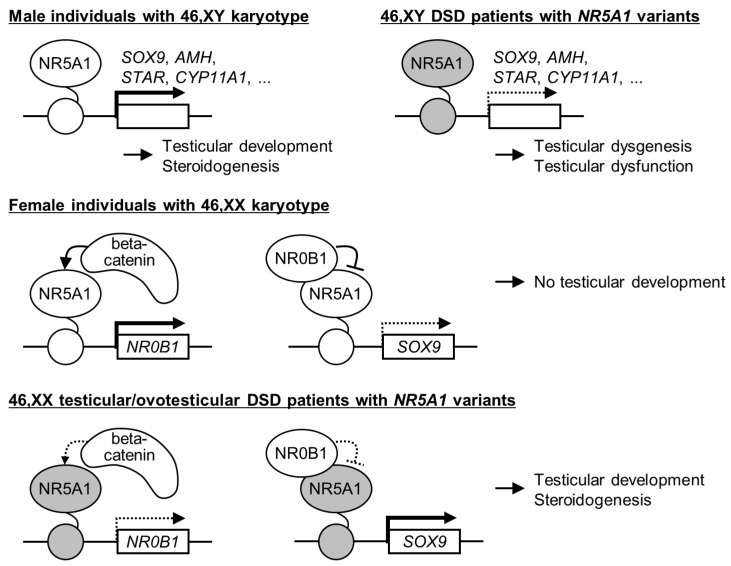
Putative mechanisms underlying 46,XY disorders/differences of sex development (DSD) and 46,XX testicular/ovotesticular DSD caused by *NR5A1* variants. White boxes depict the target genes of NR5A1. Thick arrows indicate active transcription and broken arrows depict transcriptional suppression. NR5A1 colored in gray depicts mutant proteins. NR0B1, nuclear receptor subfamily 0 group B member 1; NR5A1, nuclear receptor subfamily 5 group A member 1.

**Table 1 biomolecules-13-00691-t001:** Nuclear receptor-encoding genes that are linked to disorders/differences of sex development (DSD) at OMIM or HGMD.

HGNC-Approved Gene Symbol	Unified Nomenclature	Class	Phenotype	OMIM	HGMD	Numbers of Variants ^1^
*RARA*	NR1B1	II	46,XX DSD (Mayer-Rokitansky-Küster-Hauser syndrome)	unknown	Yes	1
*NR2F2*	NR2F2	III	46,XX testicular/ovotesticular DSD (syndromic)	Yes	Yes	3
*ESR1*	NR3A1	I	46,XX DSD (Mayer-Rokitansky-Küster-Hauser syndrome)	unknown	Yes	3
			46,XY DSD	unknown	Yes	1
*ESR2*	NR3A2	I	46,XY DSD	unknown	Yes	4
*AR*	NR3C4	I	46,XY DSD (androgen insensitivity syndrome)	Yes	Yes	647
			46,XY DSD (gonadal dysgenesis)	unknown	Yes	1
*NR5A1*	NR5A1	IV	46,XX testicular/ovotesticular DSD (nonsyndromic)	Yes	Yes	1
			46,XY DSD (gonadal dysgenesis, nonsyndromic)	Yes	Yes	232
*NR0B1*	NR0B1	unclassifiable	46,XX ovotesticular DSD (nonsyndromic)	unknown	Yes	1
			46,XY DSD (gonadal dysgenesis, syndromic and nonsyndromic)	Yes	Yes	9

^1^ Number of variants previously associated with DSD. The numbers are based on the Human Genome Mutation Database (HGMD, https://www.hgmd.cf.ac.uk/ac/index.php; accessed on 6 March 2023). HGNC, HUGO Gene Nomenclature Committee (https://www.genenames.org; accessed on 6 March 2023); OMIM, Online Mendelian Inheritance in Man (https://www.omim.org; accessed on 6 March 2023).

## Data Availability

Not applicable.

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
