# Peer review of "Nuclear Receptor Gene Variants Underlying Disorders/Differences of Sex Development through Abnormal Testicular Development"

_biomolecules, 2023, doi:10.3390/biom13040691_

Round 1
Reviewer 1 Report
This review article provides a comprehensive overview of the clinical significance of NR5A1, NR0B1, and NR2F2 variants in the etiology of DSD. It highlights the importance of gonadal development in human reproduction and emphasizes the role of aberrant gonadal development during the fetal period in causing DSD. The article describes the phenotypic variability associated with NR5A1 variants, which can lead to both 46,XY DSD and 46,XX testicular/ovotesticular DSD, and potentially contribute to digenic/oligogenic inheritances.
The article also discusses the roles of NR0B1 and NR2F2 in the etiology of DSD, including how duplications and deletions of NR0B1 can result in different forms of DSD. It notes that while NR2F2 has recently been reported as a causative gene for 46,XX testicular/ovotesticular DSD and possibly for 46,XY DSD, its precise biological role in gonadal development remains unclear.
Overall, this review provides valuable insights into the molecular networks involved in gonadal development in human fetuses, and underscores the need for further research to better understand the roles of these three nuclear receptors in sex development. There are a few questions that the authors need to address.
1. In line 352, the authors reported that the NR2F2 expression is observed in Leydig cells from seven to ten weeks of gestation but is absent at 15 weeks of gestation. However, NR2F2 is expressed in adult Leydig cells in human adult testis (Eliveld et al. 2020). Why NR2F2 is absent in fetal Leydig cells at 15 weeks of gestation but expressed in adult Leydig cells? Why the expression of NR2F2 in Leydig cells changes over time? NR2F2-expressing cells are the well-known Leydig cell progenitors. It’s strongly expressed in Leydig cell progenitors, and dimly expressed during the transition to Leydig cells.
2. In line 382, the authors should include the HSD3B1 in the steroidogenic genes.
Author Response
We are grateful for the valuable comments and suggestions. Following them, we modified our manuscript. The modified parts are highlighted in yellow.
Comment 1. In line 352, the authors reported that the NR2F2 expression is observed in Leydig cells from seven to ten weeks of gestation but is absent at 15 weeks of gestation. However, NR2F2 is expressed in adult Leydig cells in human adult testis (Eliveld et al. 2020). Why NR2F2 is absent in fetal Leydig cells at 15 weeks of gestation but expressed in adult Leydig cells? Why the expression of NR2F2 in Leydig cells changes over time? NR2F2-expressing cells are the well-known Leydig cell progenitors. It’s strongly expressed in Leydig cell progenitors, and dimly expressed during the transition to Leydig cells.
Reply: We are grateful for your important comment. We cited the article by Eliveld et al. and added sentences further discussing the NR2F2 expression pattern (lines 353–360). Specifically, we mentioned the following points.
・NR2F2 is expressed in adult Leydig cells in the adult testes.
・It is unknown whether NR2F2 is expressed in fetal Leydig cells that remain in the adult testes.
・The functions of NR2F2 likely differ between fetal and adult Leydig cells, although the precise functions of NR2F2 in the Leydig cell differentiation remain to be elucidated.
Comment 2. In line 382, the authors should include the HSD3B1 in the steroidogenic genes.
Reply: As per this comment, we added Hsd3b1 to the target genes of NR2F2 (line 389).
Reviewer 2 Report
The topic of the manuscript entitled "Nuclear Receptor Gene Variants Underlying Disorders/Differences of Sex Development through Abnormal Testicular Development" collects the knowledge of male gonadal dysgenesis and role of nuclear receptors variants. However, there is more detailed review about " Male Hypogonadism and Disordersof Sex Development" by Romina P. Grinspon, Ignacio Bergadá and Rodolfo A. Rey , which was not included in the manuscript. The figures and tables are appropriate.Conclusion are consistent with the evidence and arguments presented in the manuscript, they indicate the importance of nuclear factors in DSD. The references are appropriate.taking above togethter, I highly support the idea of publishing the review in "Biomolecules".
Author Response
We are grateful for the valuable comment. Following your comment, we modified our manuscript. The modified parts are highlighted in yellow.
Comment 1. There is more detailed review about "Male Hypogonadism and Disordersof Sex Development" by Romina P. Grinspon, Ignacio Bergadá and Rodolfo A. Rey, which was not included in the manuscript.
Reply: Thank you for introducing this important review article. We added this review article to the references (line 439) and cited it in the text (line 35).
Reviewer 3 Report
The review article by Atsushi Hattori and Maki Fukami is very well written and logically constructed. Authors carefully review the clinical significance of pathogenic variants of three nuclear receptor genes (NR5A1, NR0B1, and NR2F2) that have been reported to cause disorders of sex development in humans. The quality of English is high and the manuscript is overall well designed and concise. I only have one suggestion on a possible typo (see below).
L377 “Roles of NR0B1 in Sexual Development: Implications from Studies in Rodents” should be "Roles of NR2F2 in Sexual Development: Implications from Studies in Rodents "
Author Response
We are grateful for the valuable comment. Following your comment, we corrected our manuscript. The modified parts are highlighted in yellow.
Comment 1. L377 “Roles of NR0B1 in Sexual Development: Implications from Studies in Rodents” should be "Roles of NR2F2 in Sexual Development: Implications from Studies in Rodents"
Reply: Thank you for pointing out the mistake. We corrected the error (line 383).
Reviewer 4 Report
This review focuses on three nuclear receptors, NR5A1, NR0B1 and NR2F2, and their variants of which cause disorders/differences of sex development (DSD) through abnormal testicular development. Furthermore, it has presented a new model to investigate molecular networks in human testicular development.
It is a comprehensive review. Figures and references are adequate.
I have found this paper relevant to the field of this journal. I have only one minor comment.
Minor point:
In the title “6.3. Roles of NR0B1 in Sexual Development: Implications from Studies in Rodents” (line 377), there is a mistake. The correct title should be “6.3. Roles of NR2F2 in Sexual Development: Implications from Studies in Rodents”
I recommend this paper for acceptation after minor revision in the journal.
Author Response
We are grateful for the valuable comment. Following your comment, we corrected our manuscript. The modified parts are highlighted in yellow.
Comment 1. In the title “6.3. Roles of NR0B1 in Sexual Development: Implications from Studies in Rodents” (line 377), there is a mistake. The correct title should be “6.3. Roles of NR2F2 in Sexual Development: Implications from Studies in Rodents”
Reply: Thank you for pointing out the mistake. We corrected the error (line 383).